# Face mask use in the general population and optimal resource allocation during the COVID-19 pandemic

Colin J. Worby [1✉] & Hsiao-Han Chang [2✉]

The ongoing novel coronavirus disease (COVID-19) pandemic has already infected millions worldwide and, with no vaccine available, interventions to mitigate transmission are urgently needed. While there is broad agreement that travel restrictions and social distancing are beneficial in limiting spread, recommendations around face mask use are inconsistent. Here, we use mathematical modeling to examine the epidemiological impact of face masks, considering resource limitations and a range of supply and demand dynamics. Even with a limited protective effect, face masks can reduce total infections and deaths, and can delay the peak time of the epidemic. However, random distribution of masks is generally suboptimal; prioritized coverage of the elderly improves outcomes, while retaining resources for detected cases provides further mitigation under a range of scenarios. Face mask use, particularly for a pathogen with relatively common asymptomatic carriage, is an effective intervention strategy, while optimized distribution is important when resources are limited.

[1] Broad Institute of MIT and Harvard, 415 Main Street, Cambridge, MA 02142, USA. [2] Department of Life Science & Institute of Bioinformatics and Structural Biology, National Tsing Hua University, No. 101, Section 2, Kuang-Fu Road, Hsinchu 300044, Taiwan. ✉email: cworby@broadinstitute.org; hhchang@life.nthu.edu.tw

The rapid global spread of SARS-CoV-2 and the resulting coronavirus disease (COVID-19) pandemic has led to urgent efforts to contain and mitigate transmission, leading to significant, and widespread socioeconomic disruption[1]. By July 2020, over 10 million cases have been reported worldwide, as well as over 500,000 deaths, with ongoing spread in most parts of the world[2]. While infection is frequently asymptomatic, or associated with only mild symptoms in many people[3,4], it can cause severe and life-threatening illness in the immunocompromised and the elderly, with a case fatality ratio of over 10% in the latter group[4–6]. The rapid spread of the virus has raised concerns that healthcare systems lack sufficient resources and will be unable to bear the burden of accommodating patients suffering from COVID-19[7], resulting in significantly increased morbidity and mortality. There is an urgent need to better understand the effectiveness of potential interventions to limit the spread of the disease, especially in the context of resource limitations.

In order to mitigate the burden of infection, many countries have imposed both international and domestic travel restrictions, closed schools and nonessential businesses, and strictly limited public gatherings[8]. Such measures are designed to minimize person-to-person exposures, reducing the effective reproduction number, and thus the growth rate of the epidemic. Furthermore, individual behavior such as social distancing, self-isolation while symptomatic, handwashing and disinfecting surfaces can further mitigate transmission. Interventions such as these can offer protection (reduction in risk of infection) to susceptible individuals, and/or containment (reduction in risk of onward transmission) to infected individuals. While such measures are near universally encouraged by governments and public health departments[9], there has been limited international consensus on the use of face masks – whether surgical masks or simple reusable cloth masks—among the general public. The use of surgical masks as an infection control measure is common in East and South East Asia, and was recommended early on in the pandemic by governments in China, Hong Kong, and Taiwan for healthy persons in crowded public spaces, while masks were also recommended for symptomatic persons in Japan and Singapore[10,11]. In contrast, Western countries have been slower to encourage any adoption of masks, although there is a growing recognition that this should be part of public health policy for mitigating the spread of COVID-19[12,13]. The United States' CDC recommended cloth face coverings in April 2020[14], after many other control measures had already been implemented, while the UK government recommended cloth masks in June, limited to public transport settings only[15]. The WHO also updated their guidance in June to recommend face coverings in public, as well as medical grade masks for both high risk and symptomatic individuals, in areas with known or suspected community transmission[16]. With conflicting national guidelines and variable public compliance, self-reported mask use differs considerably between countries[17].

Some countries have seen an enormous demand for face masks from the public, with supplies being diminished and shortages reported[10]. Even with lesser public demand, the United States reported mask shortages among healthcare workers[18]. Recognizing the need for masks, several countries banned exportation of face masks[10], and the Central Epidemic Command Center in Taiwan made efforts to increase mask production in January[19]. In facing such resource shortages, it is vital that limited supplies are used effectively. Clearly, protection of staff at healthcare facilities is of critical importance, but allocating additional resources optimally among the general population can offer further benefits.

In this study, we investigate the role of face mask use and distribution among the general public during a coronavirus outbreak using mathematical modeling, in order to better understand (1) the overall reduction in infections and deaths associated with mask distribution and use, (2) how best to optimize distribution in a resource-limited setting, and (3) the role of dynamic supply and demand during an ongoing outbreak. In order to explore both the population-level effects of distributing face masks to different subpopulations, as well as capturing the supply and demand dynamics during an ongoing epidemic, we propose two models (Fig. 1). Firstly, the "resource allocation" model allows a limited number of masks to be distributed among the initial susceptible population, or allocated to symptomatic individuals while supplies are available. This allows us to compare distribution strategies in terms of final numbers of infections and deaths. Secondly, the "supply & demand" model captures dynamic mask availability, which varies in response to increased demand among the entire population as the number of reported cases increase, as well as mask production rates. We primarily assess the impact of disposable medical grade masks (i.e. a resource-limited supply; unless otherwise stated, "face masks" in this paper refer to this type) rather than homemade, reusable cloth face coverings, although we do consider implementation of both mask types in a comparison of public health policies.

Here, we demonstrate that the use of face masks among the general public is an effective strategy in mitigating transmission of SARS-CoV-2 under a range of scenarios. Nonmedical masks, when deployed widely, can also reduce total cases and deaths. We show that with a limited public supply, medical grade masks should be prioritized for vulnerable and infected individuals in order to optimize the reduction in morbidity and mortality. With no available vaccine and limited therapeutic options, face mask use is an important component of public health measures to limit the ongoing spread of COVID-19.

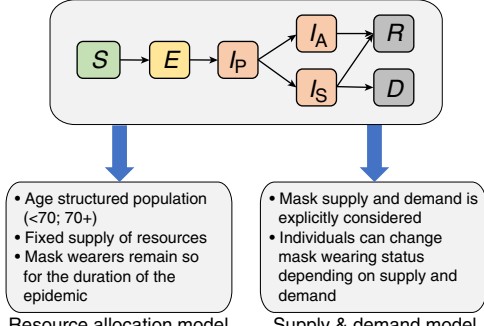

**Fig. 1 The compartmental structure common to both models.** Susceptible hosts ($S$) become exposed ($E$) and progress to presymptomatic infectious ($I_P$). Infected hosts can become either asymptomatic or mildly symptomatic ($I_A$), or symptomatic ($I_S$). Recovery ($R$) or death ($D$) follow. The resource allocation model and the supply & demand model then have unique additional features and dynamics. A schematic of the supply & demand compartmental model is shown in Supplementary Fig. 11. For a full description of each model and the specification of dynamics between compartments, see the Supplementary Methods.

## Results

**Targeted distribution of limited resources can reduce deaths**. We simulated outbreaks under a variety of parameter values associated with mask effectiveness (protection and containment) and mask supply, and identified the resulting total numbers of infections and deaths. Consistently, we found the reduction in total deaths and infections increased with mask effectiveness and availability. While immediate provision to the healthy population provided maximal impact, delayed implementation of a general

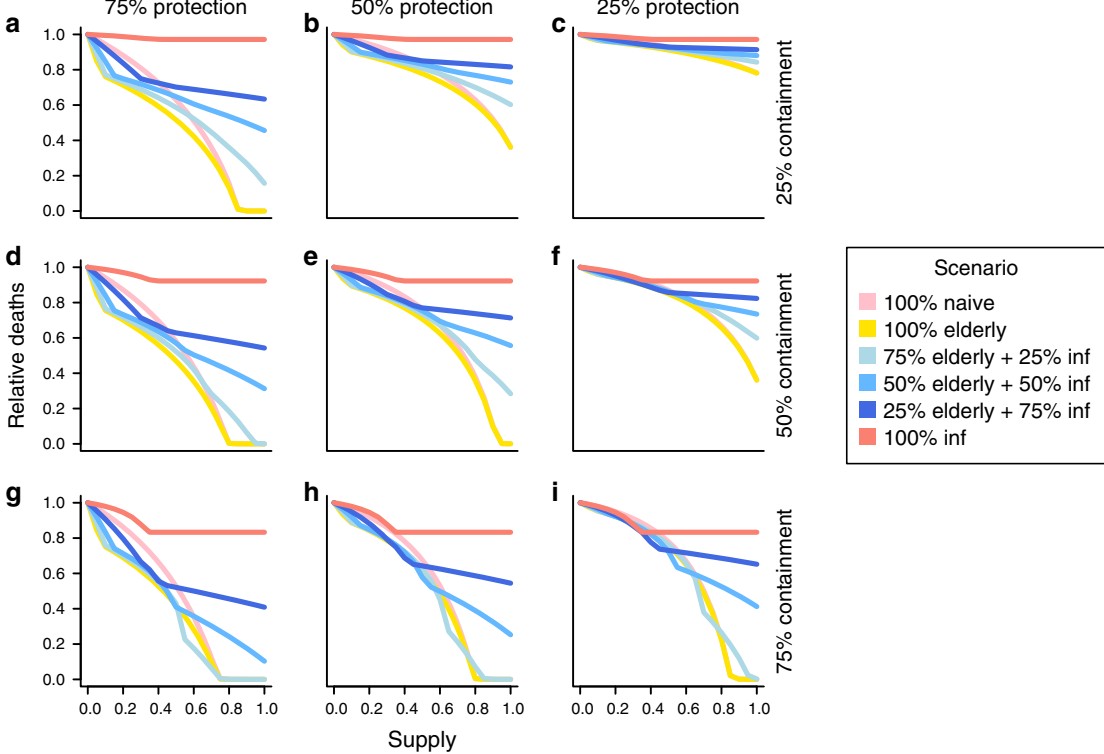

**Fig. 2 Reduction in total deaths under each distribution strategy for a range of mask availability levels.** Each panel represents mask effectiveness in terms of relative protection and containment. Containment levels of 25% **a–c**, 50% **d–f**, and 75% **g–i** are shown with varying protection levels; **c** represents the least effective mask and **g** represents the most effective mask. Masks are provided naively (pink), prioritized to the elderly (yellow), saved for detected cases (red), or balanced at different levels between healthy individuals, prioritizing the elderly, and detected cases (blue). Inflection points occur at the point where supplies are exhausted, and the outbreak continues with no new individuals adopting masks. Here, 30% of infections are assumed to be undetected. See "Methods" for further details.

mask-wearing policy could still provide reductions in total infections. The epidemic peak could be increasingly delayed with earlier adoption of mask use (Supplementary Fig. 1).

We considered a range of strategies to distribute a limited supply of masks, including (1) random distribution across the population (naive), (2) prioritized distribution to the elderly, (3) distribution to both the elderly and detected cases, and (4) distribution only to detected cases, while mask supplies last (see "Methods"). Figure 2 shows the impact of mask distribution in terms of reduced mortality under each strategy, for different levels of availability. Even limited distribution of masks offering only 25% protection and containment could result in an appreciable reduction; 10% adoption in the population could result in 5% fewer deaths (Fig. 2c). Naive distribution of masks among the general population was usually suboptimal; indeed, for a mask providing better containment than protection, this is the least optimal of the strategies we tested unless resources were plentiful (Fig. 2i). While prioritizing allocation to elderly persons only slightly reduced the total number of infections beyond that achieved with naive distribution (Supplementary Fig. 2), the number of deaths was generally considerably lower with this strategy. The benefit of prioritizing the elderly population was largest in scenarios with a limited supply of protective masks, diminishing gradually with masks offering more limited protection. With plentiful resources, the difference between prioritizing the elderly population and random distribution became limited.

It is understood that there are many other risk factors for COVID-19 morbidity and mortality in addition to advanced age[20,21]; indeed, over 20% of the population in England may be considered high risk[22]. We explored a range of dynamics in which up to 25% of the population were at elevated risk of

symptomatic illness and death (vs. the 7.6% elderly population considered in previous scenarios). The relative reduction in deaths associated with prioritized distribution increased when the high risk population was larger in most scenarios (Supplementary Fig. 3), suggesting that resource prioritization is especially important in populations with common comorbidities or with many elderly people.

**Provision only to detected cases typically has limited effect.** While it is likely that masks offer a greater degree of containment than protection, providing masks only to detected cases generally offer limited benefits, particularly when resources are plentiful (Fig. 2, red lines). Since many infections are not detected, this strategy fails to provide any containment to the large, undetected reservoir, and the benefits associated with increasing supply reach a maximum once there are sufficient resources for all detected cases. As such, this policy offers the least optimal distribution for a range of mask effectiveness parameters. By providing a mask offering intermediate levels of containment (50%) to all detected infectious cases, the number of deaths can be reduced by up to 10%, reaching this level with resources to cover 30% of the population (Fig. 2d–f). Increasing the case detection rate can further increase the benefits of this strategy (Supplementary Fig. 4).

**Optimal distribution depends on mask type and availability.** For masks offering high levels of containment, achieving a balance between providing resources to infective persons and the elderly population offers the optimal outcome in terms of total infections and deaths (Fig. 2, blue lines). These strategies offer

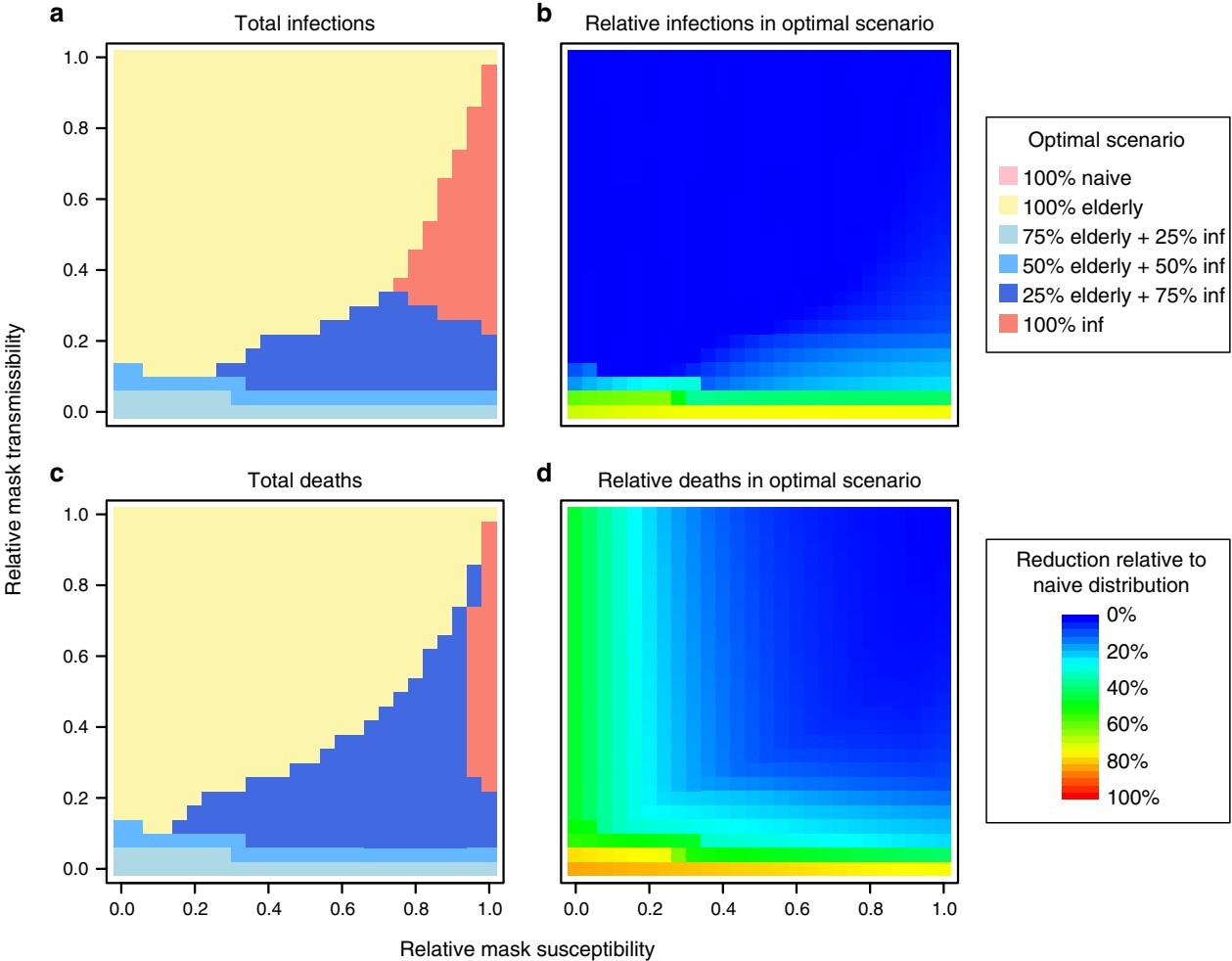

**Fig. 3 Optimal distribution of resources for different levels of mask effectiveness. a** The strategy which minimized the number of infections is indicated for each level of intervention protection and containment. With a supply of masks for 40% of the population, resources are provided under each of the strategies described in "Methods". **b** The reduction in infections under the optimal strategy is shown relative to the numbers under the naive strategy. Here we assume 30% of cases to be undetected. **c** and **d** show the equivalent plots for total deaths.

containment focused on detected cases, but also mitigate transmission from a proportion of undetected asymptomatic carriers. In addition, protection is granted to susceptible individuals, with a focus on the more vulnerable elderly population. Figure 3 shows the optimal distribution strategy over the full range of mask effectiveness parameters with resources to cover 40% of the population, as well as the corresponding reduction in total infections and deaths, relative to the naive strategy of random distribution. The optimal balance of mask distribution varied according to supply and mask effectiveness. Generally, while resources are plentiful, providing the majority of available supplies to the healthy population (prioritizing the elderly) was optimal for reducing infections and deaths.

Optimizing distribution of masks offering limited protection and containment unsurprisingly had a minimal additional reduction in morbidity and mortality beyond random distribution in the healthy population (Fig. 3b, d). However, we found that while total infections remained similar, optimizing distribution had the effect of delaying the peak time of the outbreak (e.g. Supplementary Fig. 5). In practice, this is a desirable outcome which may reduce the immediate burden on healthcare facilities.

**Panic buying prevents stockpiling and increases morbidity.** In the previous model, new mask production and ongoing supply is

not explicitly considered. Here, we investigate the role of these dynamics. We explored different scenarios of mask availability and demand by varying the parameterization of the demand function described in "Methods," as well as the rate of production of new masks. Unsurprisingly, regardless of demand dynamics, a higher production rate of masks increased availability, and therefore coverage of the population. Supplementary Fig. 6 shows the reduction in total infections given different levels of protection and mask production, highlighting that a greater number of less effective masks were required to achieve the same impact as fewer, but more effective, masks. A "panic buying" scenario, in which maximum demand for masks is attained very early in the epidemic, generally had a detrimental impact on the resulting outbreak (Supplementary Fig. 7). Unless production is ramped up during the outbreak, an inability to build a stockpile of available resources prevents people from obtaining masks readily during peak transmission (Fig. 4a, c). In contrast, a gradual increase in demand, or equivalently, a managed distribution of resources, allows for an accumulation of supplies in the early stages of the epidemic, leading to a greater availability of masks during peak transmission, and fewer overall infections (Fig. 4b, d). Specifically, a "managed demand" scenario allows a greater proportion of susceptible individuals to be covered during peak transmission, leading to fewer infections than under the panic buying scenario (Fig. 4).

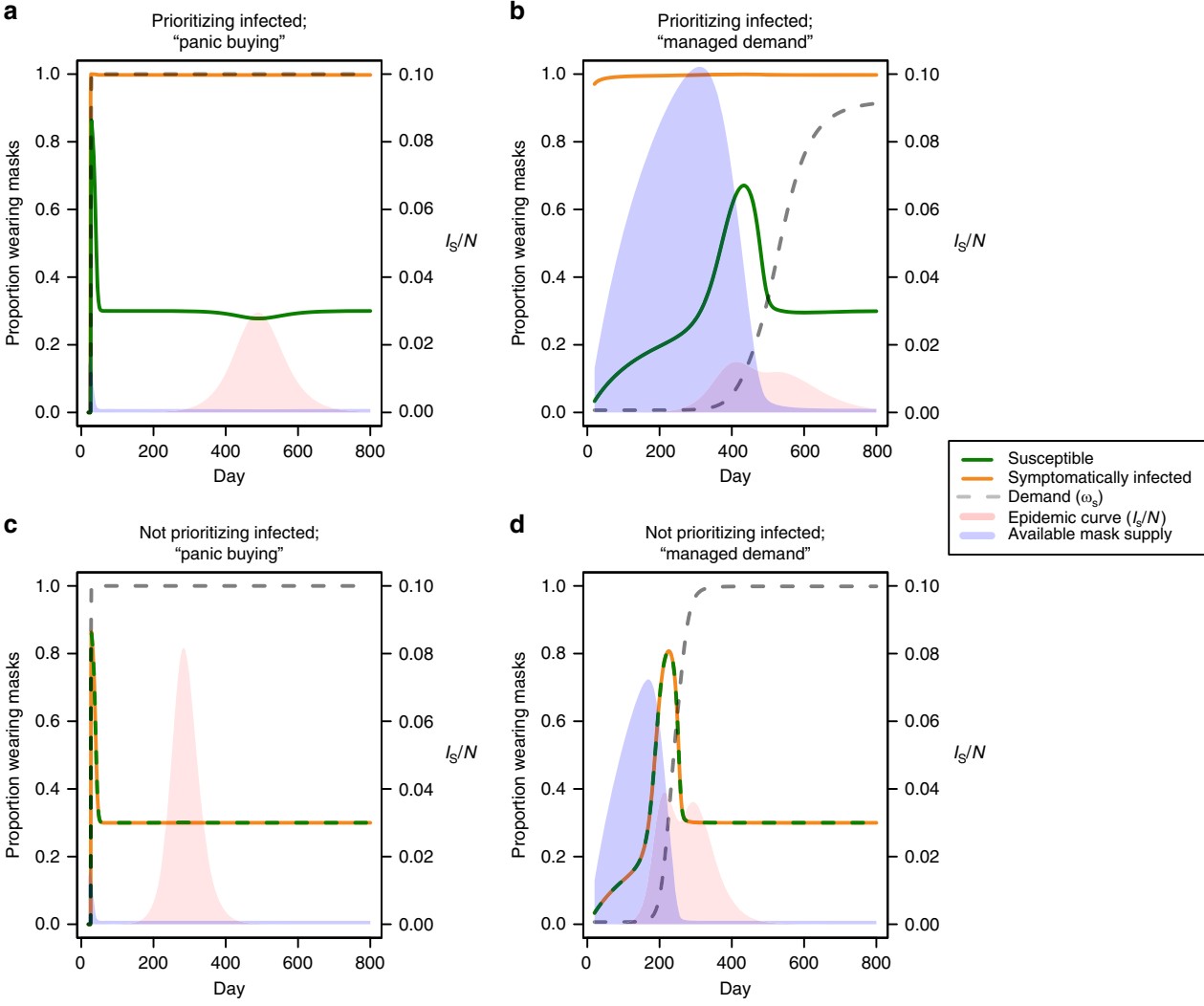

**Fig. 4 Early demand management can limit the total number of infections.** While prioritizing masks for infectious cases, epidemic curves (pink) and mask supply (blue) are shown in **a** a "panic buying" demand curve and **b** a more gradual "managed demand" curve. Demand is shown as a dashed gray line, while the proportion of the susceptible and symptomatically infected population wearing masks are shown as green and orange lines, respectively. ($k_1$, $k_2$) are (1, 100) and ($10^{-6}$, $5 \times 10^6$) for "panic buying" and managed demand, respectively (see "Methods"). The dynamics shown here are based on mask production rate, $B/N$, equal to 30%. Equivalent plots are provided showing dynamics when infectious cases are not prioritized under **c** "panic buying" and **d** "managed demand" scenarios.

With a high rate of mask production, prioritizing infectious cases allows high mask coverage to be maintained in this group during peak transmission, even in a panic buying scenario, and can reduce total infections (Fig. 4, top vs. bottom). The benefit of prioritizing masks to infectious cases gets smaller if the proportion of asymptomatic infections is higher (Supplementary Fig. 8). While building up supplies in the early phase of the epidemic can be beneficial, high levels of production were required to avoid shortages during peak demand (Supplementary Fig. 9).

**Universal face covering in public can further reduce cases**. Until now, we have only considered the distribution of resource-limited face masks (i.e. surgical masks). However, a number of countries, as well as the WHO, have introduced recommendations for the use of homemade masks, or face coverings, in public, in areas where community transmission is occurring. While the effectiveness of face coverings is likely to be limited, universal adoption would result in a reduction of $R_0$ by a factor of $r_t r_s$, where $r_t$ and $r_s$ are the relative transmission and relative

susceptibility associated with face masks, respectively. A universally adopted homemade mask offering just 5% protection and containment would thus reduce $R_0$ from 2.5 to 2.26. Adoption of universal face coverings provided a considerable reduction in total deaths (Fig. 5). This reduction was comparable to that achieved with a targeted distribution of surgical masks, even with supplies limited to 10% of the population. For a population with a universal recommendation for face covering in public (e.g. the United States post April 2020), a reduction of 3–5% in deaths may be expected; an additional targeted distribution of surgical masks to the elderly and symptomatic (e.g. WHO guidelines post June 2020) can at least double this effect.

## Discussion
During a pandemic such as COVID-19, mitigating the spread of infections is essential in the absence of a vaccine and limited critical care resources. In this study, we have shown that face mask use in the general population can have a beneficial impact in reducing the total number of infections and deaths, and that this impact naturally increases with mask effectiveness. The

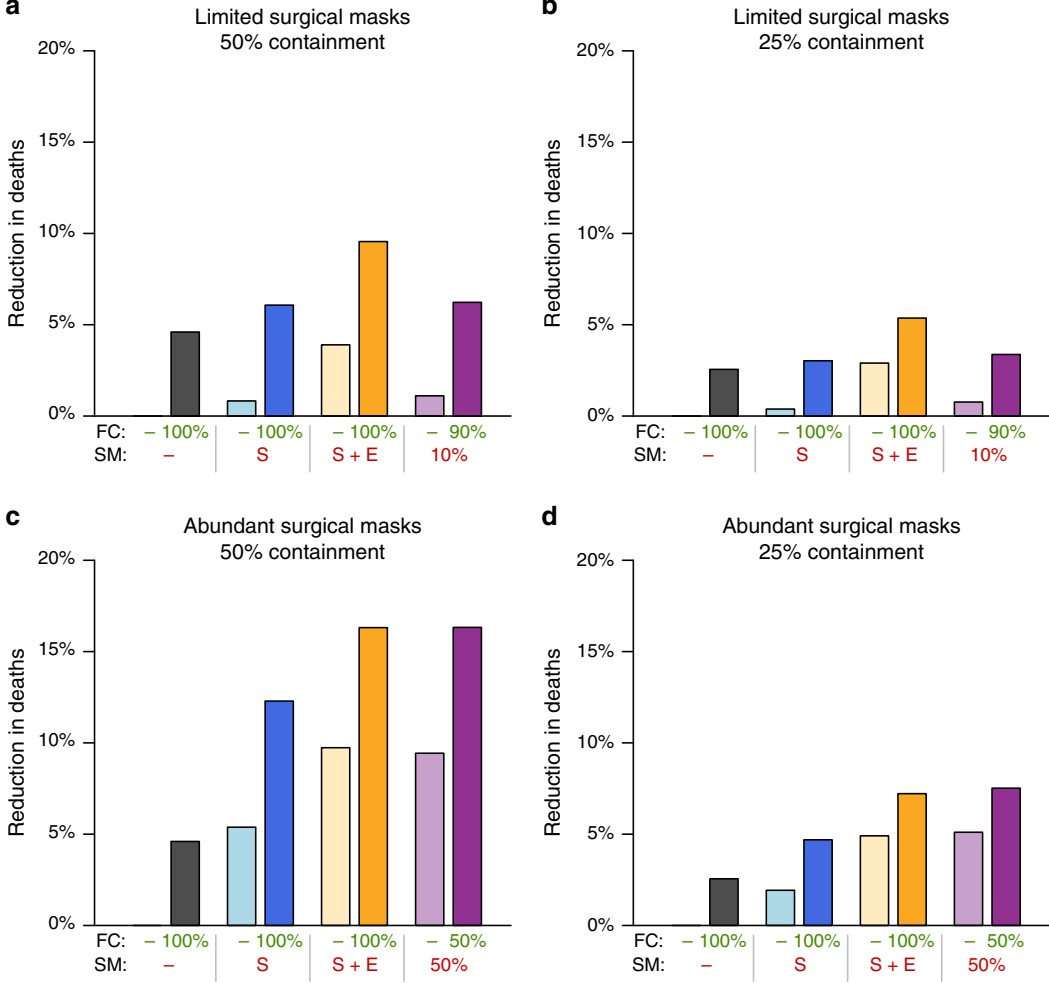

**Fig. 5 Universal face covering combined with targeted surgical mask deployment reduces total deaths.** Eight face mask policies were compared under scenarios where surgical mask (SM) supplies were limited to cover 10% **a**, **b** or 50% **c**, **d** of the population, and where surgical mask containment was high (50%, **a**, **c**) or low (25%, **b**, **d**). Available surgical masks were either not used (gray), provided to symptomatic persons (S; blue), provided to elderly and symptomatic persons (S + E; orange), or distributed randomly to the susceptible population (purple). These policies are compared with and without universal face coverings (FC) for the remaining population (darker and lighter colors, respectively). Surgical masks are assumed to confer 25% protection in all settings, and are three times more effective than face coverings.

benefits of mask deployment are apparent even with low effectiveness and limited resources. In such cases, though mask deployment may not have a large impact on total infections and deaths, indirect benefits for outbreak management are achieved by delaying the epidemic peak. Importantly, however, the overall impact of mask deployment hinges on appropriate distribution strategies. We consistently observed that the random distribution of masks throughout the general population is a suboptimal strategy. In contrast, prioritizing the elderly population, and retaining a supply of masks for identified infectious cases generally leads to a larger reduction in total infections and deaths than a naive allocation of resources.

While there remains much uncertainty around the true effectiveness of face masks—especially when factoring in differences in mask types, levels of adherence, and patterns of human behavior—there is evidence to suggest that masks can provide a measure of protection and containment for respiratory viruses. Systematic reviews have considered the reduction of transmission associated with mask wearing for respiratory viruses (odds ratio 0.32)[23], as well as SARS-CoV-2 and other betacoronaviruses (adjusted odds ratio 0.15)[24]. Cluster randomized trials involving households with diagnosed influenza cases showed significant reductions in

transmission associated with mask wearing[25], with an infection odds ratio of 0.33 when combined with handwashing[26]. Mathematical models have also suggested that the number of influenza A cases can be reduced significantly even if just a small proportion of the population wear masks[27].

Laboratory studies have also demonstrated the efficacy of masks and other fabrics as a barrier to small particles and microbes. Surgical and N95 masks limit and redirect the projection of airborne droplets[28]. Filtration efficiency, which may correlate with containment, has been estimated to be 80% for fitted surgical masks against small particles[29], or up to 96% against microbes[30]. Surgical masks were three times more effective than homemade masks, though droplet transmission from infected individuals wearing the latter was nevertheless reduced[30]. Surgical masks were estimated to significantly reduce detection of coronavirus RNA in aerosols[31], and can reduce influenza viral aerosol shedding more than threefold[32]. Generally, however, the theoretical protective effect of masks may be diminished by a number of factors. Compliance and effective use may be inadequate[25], masks may not be replaced frequently enough to prevent contamination[33], and finally, COVID-19 infection may even occur via alternative routes, such as ocular transmission[34].

In this study, we have deliberately allowed parameters in our models to vary across the full range of potential values due to the uncertainty in true mask effectiveness. A recent modeling study used effectiveness parameters of 50%, though noted the limited evidence available in setting these values[35]. Further studies are required to obtain improved estimates for mask effectiveness among the public during the COVID-19 pandemic.

Although personal protection is a leading motivator for mask wearing[36], it is generally thought that face masks are more effective in providing containment, limiting onward transmission from infectious carriers. COVID-19 is thought to have a significant proportion of mildly symptomatic or asymptomatic infections[37–39], and therefore infectious persons unaware of their status may continue to expose others. As such, even if masks offer limited personal protection, a general recommendation to wear masks in public may be particularly beneficial by containing transmission from unknowingly infectious persons.

Our models show that the more effective a mask is, the fewer masks are required to suppress an epidemic. Under a strategy in which masks are retained for infectious persons, this is particularly important. As a higher proportion of infectious persons—both symptomatic and asymptomatic (and possibly unaware)—are wearing masks offering a high level of containment, a smaller number of onward transmissions occur, requiring fewer masks to be provided for newly diagnosed individuals. As human behavior and compliance are a significant component of how effective mask use is, it is essential that public health recommendations concerning face masks in the general population occur in tandem with clear education on proper use and application, such that limited resources are used as effectively as possible[10].

While mask use can help to mitigate transmission, the supply & demand model suggests that panic buying at the very early phase of an epidemic can be detrimental, and that managing demand or increasing mask production in the early stages of the outbreak could be beneficial. In Taiwan, the government increased mask production rate and implemented such a resource management strategy in early February 2020, limiting the number of masks each person can buy per week with their National Health Insurance cards[19]. As of April 1, 2020, Taiwan was producing up to 13 million face masks a day (equivalent to $B/N \approx 0.5$), accumulating a large enough stockpile to begin exporting masks globally[40]. Since it is recommended that disposable masks should be replaced when they are soiled[41], we assumed that masks are on average worn for one day. In reality, the average lifespan of a disposable mask is likely longer due to reuse. In our model, the same dynamics are achieved by increasing mask duration or by decreasing demand. As such, a three day mask lifespan would allow a threefold reduction in mask production, resulting in equivalent epidemic dynamics, assuming no degradation in effectiveness.

Optimized deployment of resources is essential during a pandemic such as COVID-19. Our models concern the distribution of resources in the general population, under the assumption that healthcare workers and key personnel have adequate protection. If production of surgical masks can be increased such that a supply can be made available to the general population, an optimized deployment of these resources is essential. While we considered the elderly population in our model, as well as a general class of "high risk" individuals, we did not explicitly consider heterogeneity within this population. Further stratification levels for resource prioritization may lead to further reductions in infections. In addition, we did not consider heterogeneity in population mixing; in reality, there are clusters of particularly vulnerable persons (e.g., hospitals, nursing homes, prisons) which pose an elevated risk; failing to protect such communities could lead to rapid and highly localized spread. Mask provision to persons interacting with such populations (care givers, visitors, custodial staff) would likely offer greater benefits than general distribution to the public. It is likely that face mask use is also more beneficial in populations with higher contact rates. Future modeling work could consider meta-populations of different population densities to optimize resource deployment in urban vs. rural settings.

The use of face masks can be implemented simultaneously with other strategies, including social distancing, travel restrictions and self-isolation, to mitigate the spread of a pandemic disease such as COVID-19. Even during lockdown measures in which people are only rarely leaving their homes, many still face high exposure settings (e.g., conducting essential work, trips to the supermarket or drug stores) albeit less frequently. Face mask use could be a particularly important component of transmission mitigation during such activities, and widespread adoption would allow for a greater degree of interaction as more stringent lockdown measures are relaxed, while keeping the effective reproduction number below 1. Preparing an adequate supply of face masks for such a transitionary period could help to prevent a potentially costly second peak.

## Methods

**Compartmental models**. Both the resource allocation and supply & demand models share the same basic epidemic SEIRD model structure (Fig. 1) and the assumption of a closed, randomly mixing population of size $N$, similar to the model structure proposed by Anderson et al.[8]. Upon infection, susceptible individuals ($S$) enter the exposed ($E$) compartment in which a person is non-infectious, before progressing to "presymptomatic" ($I_P$) in which a person is infectious, but exhibiting no symptoms. Together, these categories represent the incubation period, after which individuals progress either to mildly symptomatic/asymptomatic ($I_A$) or symptomatic ($I_S$). Infected persons in either category can then recover ($R$), or if symptomatic, may die ($D$). Each compartment is partitioned into those wearing masks and those not. Masks reduce susceptibility to infection in healthy individuals; mask wearers' susceptibility relative to nonwearers is denoted by $r_s$, where $r_s = 0$ represents a fully protective mask. Similarly, masks are assumed to decrease transmissibility in infectious persons; a mask wearer has a relative transmissibility of $r_t$, with $r_t = 0$ representing a mask completely restricting onward transmission. For convenience, we also define the terms "protection" and "containment" as $1 - r_s$ and $1 - r_t$, respectively, and use "mask effectiveness" to refer to these properties collectively. We obtained epidemiological parameter estimates from the available literature, summarized in Supplementary Table 1. Given that estimates for the proportion of asymptomatic infections vary considerably[37–39], we allowed this parameter to vary over a plausible range; likewise, we explored a range of mask effectiveness parameters.

**Resource allocation model**. In this model, the compartments described above are further partitioned by age group (young, <70 and elderly, 70+, with the latter representing 7.6% of the population[42]). While there is currently limited evidence on the progression of infection in different age groups, we assumed that the proportion of asymptomatic infections in the elderly was half the proportion in the younger population, and that the death rate among symptomatic elderly cases was 9.7%, versus 1.3% in younger cases[43]. All symptomatic infections are assumed to be detected, while asymptomatic/mild infections are detected with a given probability $\delta$. We assume a fixed supply of masks, sufficient to protect $M_0$ persons in the population for the duration of the epidemic. Masks may be adopted by the healthy population at the start of the epidemic, while a certain proportion may be withheld for detected cases during the outbreak. Mask wearers remain as such for the duration of the epidemic. We explore a variety of distribution strategies to determine how masks may affect epidemic dynamics:

- Strategy 1: $M_0$ of the susceptible population wear masks at the start of the epidemic.
- Strategy 2: $M_0$ of the susceptible population wear masks at the start of the epidemic, with prioritized coverage of the elderly.
- Strategy 3a: $0.25M_0$ susceptible individuals wear masks at the start of the epidemic, prioritizing the elderly. Remaining masks are distributed to detected infectious individuals until supplies are diminished.
- Strategy 3b: $0.5M_0$ susceptible individuals wear masks at the start of the epidemic, prioritizing the elderly. Remaining masks are distributed to detected infectious individuals until supplies are diminished.
- Strategy 3c: $0.75M_0$ susceptible individuals wear masks at the start of the epidemic, prioritizing the elderly. Remaining masks are distributed to detected infectious individuals until supplies are diminished.

- Strategy 4: All available masks are distributed to detected infectious individuals.

For a range of mask effectiveness parameters, we identify the distribution strategy which minimizes both the number of infections and the number of deaths in the population. Full model details are described in Supplementary Methods. While we do not explicitly model individual mask use and manufacture here, this can be thought of as continuous production to provide an equilibrium number of masks which may effectively be used by up to the fraction $M_0$ of the population for the duration of the epidemic. We consider these dynamics more explicitly in the following model.

**Supply & demand model**. To understand the interplay between mask availability and disease dynamics, we model the supply and demand of face masks in a population, allowing masks to be produced at a given rate $B$, while demand may increase as the reported number of cases increases[10,44]. In this model, we allow for movement between mask-wearing and nonmask-wearing status, depending on availability and demand. Masks must be continually acquired to remain a mask wearer. We assume that the mask is worn on average for $\mu$ days before requiring replacement, and the rate of nonwearers acquiring masks depends on both demand ($\omega_A$ for healthy and asymptomatic individuals, or $\omega_S$ for symptomatic individuals) and current supply ($M/N$, the proportion of mask in the overall population). We assume that demand for masks increases with the number of reported cases, up to a certain plateau, and as such model the relationship between the demand and the number of symptomatic infections in the following sigmoidal function, with shape parameters $k_1$ and $k_2$:

$$\omega_A = \frac{1}{1 + e^{-k_1(I_S - k_2)}}. \tag{1}$$

The range of $\omega_A$ is [0, 1], $k_1$ represents the rate of demand increase, and $k_2$ represents the timing of demand, defined as the number of reported cases when half the population seeks face masks (i.e., higher $k_2$ means mask demand increases later in the outbreak). This parameterization allows us to explore different demand dynamics, for example, "panic buying" (high $k_1$), delayed response to epidemic threat (high $k_2$), a gradual increase in demand ("managed demand") (low $k_1$, high $k_2$) (Supplementary Fig. 10).

**Comparison of face mask policies**. Guidance around the use of face masks varies widely, and several countries in Europe and North America recommend, or mandate, the use of homemade face coverings in public. In order to compare such recommendations to the deployment of resource-limited surgical masks, we ran a further set of simulations under the resource allocation model, incorporating universal face covering. Specifically, we considered (1) no surgical mask use, (2) provision of surgical masks to symptomatic cases, (3) provision of surgical masks to the elderly population as first priority and then symptomatic cases if available, and (4) random surgical mask distribution. Each policy was considered both with and without universal face covering for the remainder of the population. We compared these eight policies for different levels of surgical mask availability and effectiveness. Surgical masks were assumed to be three times more effective than face coverings[30], both in terms of protection and containment.

**Reporting summary**. Further information on research design is available in the Nature Research Reporting Summary linked to this article.

## Data availability
No datasets were generated or analyzed in this modeling study. Parameter values for models were obtained from the literature as described in Supplementary Table 1.

## Code availability
All models and analyses were run in R[45], using the deSolve package[46]. Code to run both models described in this paper is available at github (https://github.com/hhc-lab/mask_covid-19).

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

## Acknowledgements
The authors wish to acknowledge Lee Kennedy-Shaffer and Yu-Lun Liu for helpful discussions. This study was supported by the Ministry of Science and Technology in Taiwan (MOST 109-2636-B-007-006). C.J.W. received support from the National Institute of Allergy and Infectious Diseases (grant number U19AI110818). The funders had no role in preparation of the manuscript.

## Author contributions
H-.H.C. conceptualized the study. C.J.W. and H-.H.C. designed and implemented models, performed analyses, interpreted results, and wrote the manuscript.

## Competing interests
The authors declare no competing interests.
