## [Peer Review File · Nature Communications]

REVIEWER COMMENTS

Reviewer #1 (Remarks to the Author):

This paper by Worby and Chang examines how effective mask use can be at reducing cases and deaths from COVID-19 under different scenarios for the efficacy of masks and preventing transmission from infected wearers or preventing susceptible wearers from catching infection, and under different assumptions about mask availability. I think this is a very important and timely paper and the analysis is very thorough. Mask use is hotly debated in many countries: while mask use is common in eastern Asian countries, it is only recently being adopted in some US regions or European nations, and is relatively uncommon in many regions. Despite the fact that high-quality surgical masks are extremely cheap to manufacture, nearly all countries suffer from mask shortages, and confusing public health messaging about the efficacy of masks for different has limited their use.

There are many interesting and conclusions of this paper that wouldn't have been obvious without a detailed model to compare scenarios, and I think they will be helpful for policy makers and public health officials to make the case for mask use.

For example, re Figure 3 results, I found it surprising that giving masks to uninfected ppl ("naive") was only the optimal strategy when mask supply was high - even when masks much better at protection than containment. It was interesting that this strategy is the only one that really switches order with other strategies as supply changes. It was also surprising that it's never good to give mask to only infected people - even when masks are much better at containment than protection! I also wouldn't have expected that the relative ordering of strategies doesn't really change with the different % containment and % protective efficacy or even really with supply. If there are significant differences between the strategies, its generally always better to prioritize the elderly (even though you aren't assuming any preferential mixing by age). Re Figure 4 results, I found it interesting that if you had to choose between a mask with 100% reduction in transmissibility + 0% reduction in susceptibility, vs 100% reduction in susceptibility but 0% reduction in transmissibility, at 40% mask availability, and you used the optimal allocation strategy in each case, you would get much more reduction in infections and deaths for a mask that reduced transmissibility.... even with the assumption of imperfect detection.

I would definitely recommend this paper for publication. All the major and minor claims are supported by analysis and the model is so clearly explained and documented that it would be very easy for anyone to reproduce. This is the first paper I have seen to examine the effect of mask use on COVID-19 spread.

I just have a few minor comments:

- * Figure 2: Could you add another figure panel which shows final % dead and maybe also final % infected for these different scenarios? It's nice to see the timecourse but hard to mentally relate to overall effect

- * Figure 4: Would be helpful to have a legend for the figure saying what each color line means. It's really cumbersome to have to read the caption text to try to figure this out and makes reading an already busy figure that much harder.

- * Re parameters: The authors should cite a primary source for parameter values, not a (quite old) modeling paper that doesn't give much details about its own methods or parameter sources (i.e. reference 8, Anderson et al Lancet). Most parameters seemed reasonable, except the length of the true incubation period (E) seemed very short at 1 day. I don't think that's what most other papers have concluded when comparing period without symptoms to serial interval. Also, why do you assume masks are worn for only 1 day? I know you are assuming "disposable" masks but even disposable masks can be worn way more than this, especially for people who are not doing manual labor (different for healthcare workers of course).

Reviewer #2 (Remarks to the Author):

Using SEIRD modelling (resource allocation and supply and demand), this study demonstrates the wearing face mask's beneficial impact in terms of number of infections and deaths. The authors evaluate the prioritizations of face mask on elderly groups giving limited resources. This is a nice and timely modelling study assessing the impact of face mask strategies.

This study appears based on Taiwan scenario (elderly population size, mask production etc), where it has been stockpiled large number of mask even before the pandemic. However, many countries/regions are still suffer from shortage in mask supply, while some of these countries recommend cloth face covering. My suggestion would be consider to extend to wider scenarios based on current face mask recommendations in other countries/regions, especially compulsory face covering policy that have been implemented in the US and some European countries. I specify my suggestions in details as follows:

1. Consider to evaluate other practical scenarios and ongoing strategies in resource allocation model, i.e., mandatory cloth face covering policy, and potential impact by using different type of mask. It would be very interesting to examine the impact on number of infections and deaths under this policy, considering the type of mask/face covering by varying the "protection" and "containment" and supply. It is reasonable to expect cloths face covering having lower "protection" and "containment" but universal available throughout the epidemic.

Taking the US as an example, it would be implicative to compare scenarios between 1) symptomatic population wearing face mask (previous recommendation), 2) 1) plus mandatory face covering policy among every susceptible population and asymptomatic individuals (current recommendation), 3) 2) plus prioritizing surgical mask to exposed (asymptomatic), elderly and vulnerable population when supply permits, 4) universal surgical face mask when supply permits. Modelling the number of infections and deaths averted would allow the assessment of impact of updated policy and recommendations.

2. It would be also interesting to incorporating the uncertainty of effectiveness by different type of cloths mask (tea tower, cotton etc), allowing heterogeneity on "protection" and "containment" within population (<https://pubmed.ncbi.nlm.nih.gov/24229526/>).

3. Incorporate high-risk population of which the proportion might be over 20% in some countries and there might be large impact on prioritising interventions ([https://doi.org/10.1016/S0140-6736\(20\)30854-0](https://doi.org/10.1016/S0140-6736(20)30854-0)).

4. Besides, as the pandemic goes fast, the numbers of cases and deaths in Background need to be updated. There is a crucial paper on effectiveness of face mask that might be helpful to cite: [https://doi.org/10.1016/S0140-6736\(20\)31142-9](https://doi.org/10.1016/S0140-6736(20)31142-9)

5. Figure 2. Not sure what the red circles represent. Might be helpful to summarise the numbers (e.g. how many days delayed, total size of epidemic etc) in a table, as it is a bit hard to tell from the figure.

Detailed responses to reviewers

Queries/critiques are numbered and in blue Cambria italic font. Responses follow in black Cambria font. Revised prose from within the manuscript is in black Times Roman.

Reviewer #1 (Remarks to the Author):

This paper by Worby and Chang examines how effective mask use can be at reducing cases and deaths from COVID-19 under different scenarios for the efficacy of masks and preventing transmission from infected wearers or preventing susceptible wearers from catching infection, and under different assumptions about mask availability. I think this is a very important and timely paper and the analysis is very thorough. Mask use is hotly debated in many countries: while mask use is common in eastern Asian countries, it is only recently being adopted in some US regions or European nations, and is relatively uncommon in many regions. Despite the fact that high-quality surgical masks are extremely cheap to manufacture, nearly all countries suffer from mask shortages, and confusing public health messaging about the efficacy of masks for different has limited their use.

There are many interesting and conclusions of this paper that wouldn't have been obvious without a detailed model to compare scenarios, and I think they will be helpful for policy makers and public health officials to make the case for mask use.

For example, re Figure 3 results, I found it surprising that giving masks to uninfected ppl ("naive") was only the optimal strategy when mask supply was high - even when masks much better at protection than containment. It was interesting that this strategy is the only one that really switches order with other strategies as supply changes. It was also surprising that it's never good to give mask to only infected people - even when masks are much better at containment than protection! I also wouldn't have expected that the relative ordering of strategies doesn't really change with the different % containment and % protective efficacy or even really with supply. If there are significant differences between the strategies, its generally always better to prioritize the elderly (even though you aren't assuming any preferential mixing by age). Re Figure 4 results, I found it interesting that if you had to choose between a mask with 100% reduction in transmissibility + 0% reduction in susceptibility, vs 100% reduction in susceptibility but 0% reduction in transmissibility, at 40% mask availability, and you used the optimal allocation strategy in each case, you would get much more reduction in infections and deaths for a mask that reduced transmissibility.... even with the assumption of imperfect detection.

I would definitely recommend this paper for publication. All the major and minor claims are supported by analysis and the model is so clearly explained and documented that it would be very easy for anyone to reproduce. This is the first paper I have seen to examine the effect of mask use on COVID-19 spread.

Response: We are very grateful for the reviewer's supportive comments. We agree that findings from our model highlight dynamics which might not be obvious at first

consideration, and hope that our contribution can provide additional insight into the prioritization of limited resources during the pandemic.

I just have a few minor comments:

1. Figure 2: Could you add another figure panel which shows final % dead and maybe also final % infected for these different scenarios? It's nice to see the timecourse but hard to mentally relate to overall effect.

Response: We have updated the former Figure 2 to include additional panels showing reductions in deaths and delay in epidemic time. In consideration of the additional simulations incorporated into the paper (see below), we have decided to make the resulting figure part of the supplementary material – it is now Figure S2.

2. Figure 4: Would be helpful to have a legend for the figure saying what each color line means. It's really cumbersome to have to read the caption text to try to figure this out and makes reading an already busy figure that much harder.

Response: Thank you for your suggestion. We have added a legend for the figure explaining what each color means.

3. Re parameters: The authors should cite a primary source for parameter values, not a (quite old) modeling paper that doesn't give much details about its own methods or parameter sources (i.e. reference 8, Anderson et al Lancet). Most parameters seemed reasonable, except the length of the true incubation period (E) seemed very short at 1 day. I don't think that's what most other papers have concluded when comparing period without symptoms to serial interval. Also, why do you assume masks are worn for only 1 day? I know you are assuming "disposable" masks but even disposable masks can be worn way more than this, especially for people who are not doing manual labor (different for healthcare workers of course).

Response: We note here that our model parameterization was unclear in the previous manuscript - apologies. Incubation period in our model includes both time from exposed to pre-symptomatic infections (=1 day) and time from pre-symptomatic to symptomatic infections (=5 days) and has an average of 6 days. We now included citations for the length of incubation period in the Supplementary Table, and added a sentence in Supplementary Text as follows:

“The average incubation period ($1/\alpha_1+1/\alpha_2$) is assumed to be 6 days [45-47]. ”

Additionally we clarified the text in the Methods:

“Upon infection, susceptible individuals (S) enter the exposed (E) compartment in which a person is non-infectious, before progressing to ‘pre-symptomatic’ (I_P) in which a person is infectious, but exhibiting no symptoms. Together, these categories represent the incubation period,…”

We originally assumed masks are worn for 1 day because this is the recommendation from the Taiwan CDC. We agree that masks can potentially last longer, and performed sensitivity analysis for 3 and 7 days. Increasing mask duration is equivalent to decreasing demand. We added few sentences on page 9 to discuss this point as follows:

“Since it is recommended that disposable masks should be replaced when they are soiled [43, 44], we assumed that masks are on average worn for one day. In reality, the average lifespan of a disposable mask is likely longer due to reuse. In our model, the same dynamics are achieved by increasing mask duration or by decreasing demand. As such, a three day mask lifespan would allow a threefold reduction in mask production, resulting in equivalent epidemic dynamics, assuming no degradation in effectiveness.”

Reviewer #2 (Remarks to the Author):

Using SEIRD modelling (resource allocation and supply and demand), this study demonstrates the wearing face mask's beneficial impact in terms of number of infections and deaths. The authors evaluate the prioritizations of face mask on elderly groups giving limited resources. This is a nice and timely modelling study assessing the impact of face mask strategies.

This study appears based on Taiwan scenario (elderly population size, mask production etc), where it has been stockpiled large number of mask even before the pandemic. However, many countries/regions are still suffer from shortage in mask supply, while some of these countries recommend cloth face covering. My suggestion would be consider to extend to wider scenarios based on current face mask recommendations in other countries/regions, especially compulsory face covering policy that have been implemented in the US and some European countries. I specify my suggestions in details as follows:

1. Consider to evaluate other practical scenarios and ongoing strategies in resource allocation model, i.e., mandatory cloth face covering policy, and potential impact by using different type of mask. It would be very interesting to examine the impact on number of infections and deaths under this policy, considering the type of mask/face covering by varying the “protection” and “containment” and supply. It is reasonable to expect cloths face covering having lower “protection” and “containment” but universal available throughout the epidemic.

Taking the US as an example, it would be implicative to compare scenarios between 1) symptomatic population wearing face mask (previous recommendation), 2) 1) plus mandatory face covering policy among every susceptible population and asymptomatic individuals (current recommendation), 3) 2) plus prioritizing surgical mask to exposed (asymptomatic), elderly and vulnerable population when supply permits, 4) universal surgical face mask when supply permits. Modelling the number of infections and deaths averted would allow the assessment of impact of updated policy and recommendations.

Response: Many thanks for this excellent suggestion – we have incorporated additional simulations based on face covering policies, and believe that this makes a valuable addition

to the paper. We considered a range of surgical mask recommendations both with and without universal face coverings, for different mask availability and containment. On the basis of the paper below, we assumed face coverings were three times less effective than surgical masks. We created Figure 5 to highlight our findings from these analyses, and added the methods section “*Comparison of face mask policies*” and the results section “*Universal adoption of face coverings in public can further reduce cases*” as follows:

In Methods:

Comparison of face mask policies

Guidance around the use of face masks varies widely, and several countries in Europe and North America recommend, or mandate, the use of homemade face coverings in public. In order to compare such recommendations to the deployment of resource-limited surgical masks, we ran a further set of simulations under the resource allocation model, incorporating universal face covering. Specifically, we considered (i) no surgical mask use, (ii) provision of surgical masks to symptomatic cases, (iii) provision of surgical masks to the elderly population as first priority and then symptomatic cases if available, and (iv) random surgical mask distribution. Each policy was considered both with and without universal face covering for the remainder of the population. We compared these eight policies for different levels of surgical mask availability and effectiveness. Surgical masks were assumed to be three times more effective than face coverings [23], both in terms of protection and containment.”

In Results:

Universal adoption of face coverings in public can further reduce cases

Until now, we have only considered the distribution of resource-limited face masks (i.e. surgical masks). However, a number of countries, as well as the WHO, have introduced recommendations for the use of homemade masks, or face coverings, in public. While the effectiveness of face coverings is likely to be limited, universal adoption would result in a reduction of R_0 by a factor of $r_f r_s$; a universally-adopted homemade mask offering just 5% protection and containment would thus reduce R_0 from 2.5 to 2.26. Adoption of universal face coverings provided a considerable reduction in total deaths (Figure 5). This reduction was comparable to that achieved with a targeted distribution of surgical masks, even with supplies limited to 10% of the population. For a population with a universal recommendation for face covering in public (e.g. the USA post April 2020), a reduction of 3-5% in deaths may be expected; an additional targeted distribution of surgical masks to the elderly and symptomatic (e.g. WHO guidelines post June 2020) can double this effect.

Figure 5. Universal face covering provides additional benefits combined with targeted surgical mask deployment. Eight face mask policies were compared under scenarios where surgical mask (SM) supplies were limited to cover 10% (top) or 50% (bottom) of the population, and where surgical mask containment was high (50%, left) or low (25%, right). Available surgical masks were either not used (gray), provided to symptomatic persons (S; blue), provided to elderly and symptomatic persons (S+E; green), or distributed randomly to the susceptible population (red). These policies are compared with and without universal face coverings (FC) for the remaining population (darker and lighter colors, respectively). Surgical masks are assumed to confer 25% protection in all settings, and are three times more effective than face coverings.

2. It would be also interesting to incorporating the uncertainty of effectiveness by different type of cloths mask (tea tower, cotton etc), allowing heterogeneity on “protection” and “containment” within population (<https://pubmed.ncbi.nlm.nih.gov/24229526/>)

In our simulations above, we evaluated two levels of cloth face covering containment, equivalent to 8.3% and 16%, in tandem with the surgical mask covering of 25% and 50% containment respectively. Our model does not allow heterogeneity within a given population readily, and we suspect that variation across the relatively low range of

homemade face mask effectiveness parameters would not impact overall results significantly.

3. Incorporate high-risk population of which the proportion might be over 20% in some countries and there might be large impact on prioritising interventions ([https://doi.org/10.1016/S0140-6736\(20\)30854-0](https://doi.org/10.1016/S0140-6736(20)30854-0)).

Many thanks for this suggestion. We have performed additional analyses with a high risk population ranging up to 25% (vs. the 8% elderly population previously under consideration). We found that the relative reduction in deaths associated with prioritized distribution to the high risk population increased when this population increased – suggesting an increased importance of prioritization for populations with more high risk individuals. We have added text to this effect in the results section, and added Figure S4 as follows:

“It is understood that there are many other risk factors for COVID-19 morbidity and mortality in addition to advanced age, including cardiovascular disease [38], obesity [39], and diabetes [40]; indeed, over 20% of the population in England may be considered high risk [41]. We explored a range of dynamics in which up to 25% of the population were at elevated risk of symptomatic illness and death (vs. the 7.6% elderly population considered in previous scenarios). The relative reduction in deaths associated with prioritized distribution to the high risk population increased when this population represented a larger proportion for intermediate levels of mask availability (Figure S4), suggesting that resource prioritization is especially important in such cases.

Figure S4. Prioritized distribution to high risk individuals is increasingly optimal for larger high risk populations. For a mask providing 50% containment and 75% protection, we explored dynamics in populations with different ‘high-risk’ communities and mask supplies. For all scenarios considered, prioritized distribution to the elderly was the optimal strategy. This strategy was associated with greater reductions in deaths relative to the random distribution

strategy, as shown by the colors. The dashed line indicates where mask supply is equal to the size of the high risk population.

4. Besides, as the pandemic goes fast, the numbers of cases and deaths in Background need to be updated. There is a crucial paper on effectiveness of face mask that might be helpful to cite: [https://doi.org/10.1016/S0140-6736\(20\)31142-9](https://doi.org/10.1016/S0140-6736(20)31142-9)

Response: We have now updated the number of cases and deaths in Background as follows:

“By July 2020, over 10 million cases have been reported worldwide, as well as over 500,000 deaths, with ongoing spread in most parts of the world [2].”

Thanks for pointing us to this interesting paper. We cited it in Background as follows:

“A recent meta-analysis of SARS-CoV-2 and other betacoronaviruses found face mask use could significantly reduce risk of infection [18]”

5. Figure 2. Not sure what the red circles represent. Might be helpful to summarise the numbers (e.g. how many days delayed, total size of epidemic etc) in a table, as it is a bit hard to tell from the figure.

We have remade Figure 2 on the basis of this comment and that of Reviewer 1. It is now Figure S2.

REVIEWERS' COMMENTS:

Reviewer #1 (Remarks to the Author):

The authors have addressed all of my comments (which were only minor) and seems to have done significant extra analysis in response to the other reviewers concerns. The paper is even nicer now than before and I would highly recommend publication.

I only have one extra minor suggestion which the authors can address at their own discretion. I understand that there is uncertainty in the % efficacy for containment and protection, and so in simulations they use a few different values, but I really wanted to see more discussion (in the Introduction or Discussion or both) of the actual numerical values for efficacy that have been estimated in different studies. I think that in order for this study to have impact among non-scientists, it is necessary to convince people that we have at least idea what these numbers are!

Reviewer #2 (Remarks to the Author):

The previous comments have been well addressed and I'm happy with responses and updates.

Response to Reviewers

Reviewer #1 (Remarks to the Author):

The authors have addressed all of my comments (which were only minor) and seems to have done significant extra analysis in response to the other reviewers concerns. The paper is even nicer now than before and I would highly recommend publication.

I only have one extra minor suggestion which the authors can address at their own discretion. I understand that there is uncertainty in the % efficacy for containment and protection, and so in simulations they use a few different values, but I really wanted to see more discussion (in the Introduction or Discussion or both) of the actual numerical values for efficacy that have been estimated in different studies. I think that in order for this study to have impact among non-scientists, it is necessary to convince people that we have at least idea what these numbers are!

Many thanks to the reviewer for this feedback, and all suggestions from earlier versions of the manuscript. We deliberately use a wide range of values to explore potential mask effectiveness parameters, but completely agree that it would be useful to have more grounding in the values and uncertainty that have been estimated previously in the literature. We hope that better estimates can be obtained in the coming months. We provided more numerical values based on previous studies in the following paragraphs in the discussion:

Systematic reviews have considered the reduction of transmission associated with mask wearing for respiratory viruses (odds ratio 0.32)²⁴, as well as SARS-CoV-2 and other betacoronaviruses (adjusted odds ratio 0.15)²⁵. Cluster randomized trials involving households with diagnosed influenza cases showed significant reductions in transmission associated with mask wearing²⁶, with an infection odds ratio of 0.33 when combined with handwashing²⁷. Mathematical models have also suggested that the number of influenza A cases can be reduced significantly even if just a small proportion of the population wear masks²⁸.

Laboratory studies have also demonstrated the efficacy of masks and other fabrics as a barrier to small particles and microbes. Surgical and N95 masks limit and redirect the projection of airborne droplets²⁹. Filtration efficiency, which may correlate with containment, has been estimated to be 80% for fitted surgical masks against small particles³⁰, or up to 96% against microbes³¹. Surgical masks were three times more effective than homemade masks, though droplet transmission from infected individuals wearing the latter was nevertheless reduced³¹. Surgical masks were estimated to significantly reduce detection of coronavirus RNA in aerosols³², and can reduce influenza viral aerosol shedding more than threefold³³. Generally however, the theoretical protective effect of masks may be diminished by a number of factors. Compliance and effective use may be inadequate²⁶, masks may not be replaced frequently enough to prevent contamination³⁴, and finally, COVID-19 infection may even occur via alternative routes, such as ocular transmission³⁵.

In this study, we have deliberately allowed parameters in our models to vary across the full range of potential values due to the uncertainty in true mask effectiveness. A recent modeling study used effectiveness parameters of 50%, though noted the limited evidence available in setting these values³⁶. Further studies are required to obtain improved estimates for mask effectiveness among the public during the COVID-19 pandemic.

Reviewer #2 (Remarks to the Author):

The previous comments have been well addressed and I'm happy with responses and updates.

We thank the reviewer for the positive feedback and constructive suggestions raised during the review process.